# Long Distance Propagation of 162-MHz Shipping Information Links Associated with Sporadic-E

Alex T. Chartier[1], Thomas R. Hanley[1], Daniel J. Emmons[2]

[1] Johns Hopkins University Applied Physics Laboratory, Laurel, MD 20723, United States of America

[2] Air Force Institute of Technology, Wright-Patterson AFB, OH 45433, United States of America

*Correspondence to*: Alex T. Chartier (alex.chartier@jhuapl.edu)

**Abstract.** Sporadic-E layers form in the daytime midlatitude ionosphere as a result of wind shears in the mesosphere-lower thermosphere compressing metallic ions of meteoric origin into dense, narrow sheets extending over hundreds or thousands of kilometers spatially. These layers are poorly observed, being too narrow to be properly resolved by incoherent scatter radar or path-integrated total electron content measurements. Sporadic-E layer peak densities can be resolved by ionosondes and by rocket-borne Langmuir probes, but these techniques have major limitations in terms of spatial and temporal coverage, and (for many ionosondes) maximum density resolution. As a result, the density, occurrence and spatial extent of sporadic-E layers are not well constrained by observations. The maximum density of sporadic-E is widely believed to be around 5-10 x $10^{11}$ el. $m^{-3}$ $N_mE_s$ (equivalent to $6 - 9$ MHz $f_oE_s$), though there are a few isolated reports of layers extending beyond 20 MHz (Chandra and Rastogi, 1975; Maeda and Heki, 2014). Here, we identify sporadic-E layers using a huge database of 29 million 162-MHz Automatic Identification System (AIS) shipping transmissions collected over three days by a United States Coast Guard (USCG) terrestrial monitoring network in the eastern United States and Puerto Rico. Within this dataset, most (>99%) links are explained by line-of-sight, surface-wave and tropospheric propagation, but a small population cannot be explained by these mechanisms. In total, 6677 signals were identified from ships located over 1000 km from the ground stations between 13 and 14 July 2021, and almost no long-distance links were received at night or at any time on 15 July. This coincides with intense (saturated) sporadic-E in collocated ionosondes and in satellite radio occultation data. The density of these layers might exceed 27 MHz $f_oE_s$, or 9x$10^{12}$ el. $m^3$ $N_mE_s$. AIS transmissions potentially provide an excellent means of identifying dense sporadic-E layers globally.

## 1 Introduction

Automatic Identification System (AIS) transponders are designed to provide vessel position, identification and other information to other ships and to coastal authorities (e.g. IMO, 2022). The system operates using Very High Frequency (VHF) radio transmissions on two 25 kHz channels close to 162 MHz. AIS is required to be used on large ships, typically at the 12.5 Watt level. Many smaller vessels, including recreational boats, are also fitted with low power (~2 Watts) or passive AIS systems. The United States Coast Guard (USCG) operates a network of land-based AIS monitors, and provided three days of data from stations in the eastern USA and Puerto Rico for this study (13 – 15 July 2021) along with satellite-received data used as an independent point-of-reference. The exact station locations are not disclosed by request of the data provider.

Occasionally, signals are identified at long distances of 1000 km or more. This long distance propagation is surprising since signals at 162 MHz would typically be expected to pass through the ionosphere to space, rather than reflecting off and back to Earth as skywaves. VHF signals do propagate over the horizon as surface waves and through tropospheric ducting (Ames et al., 1955) but the distances are typically limited to a few hundred kilometers or less. We note that very long distance propagation of VHF signals has been reported in amateur radio databases. The "More Miles on VHF" database (https://mmmonvhf.de/odx.php) reports links up to 4966 km. All the top 100 links listed in that dataset occurred between May and August, which is coincident with the sporadic-E season in the northern hemisphere.

This study evaluates the possibility that extremely dense, low-altitude ionospheric layers (known as sporadic-E) could provide a skywave propagation path that would explain the long-range USCG AIS observations. Sporadic-E layers were first identified using ionosondes (e.g Thomas and Smith, 1959). These layers are known to occur frequently, especially at mid-latitudes during the daytime in summer (Wu et al., 2005; Chu et al, 2014; Arras and Wickert, 2018), with the cause believed to be redistribution of existing plasma into thin, dense layers by wind shears (see reviews by Whitehead, 1970; Mathews, 1998; Haldoupis, 2011 for more details). The process may be aided by the presence of long-lived metallic ions deposited in the lower ionosphere by meteors (e.g. Maruyama et al., 2008). Recently, Yamazaki et al. (2022) presented convincing evidence linking sporadic-E to zonal wind shears using ICON/MIGHTI interferometer wind profile data and COSMIC-2/RO retrieved electron density profiles. Deacon et al. (2022) have linked sporadic-E to long-distance amateur radio propagation reports on frequencies up to 70 MHz, while amateur groups themselves routinely map sporadic-E (e.g. Sampol, retrieved 2022). There are some observations of extremely high sporadic-E critical frequency ($f_oEs$), for example Chandra and Rastogi (1975), Maeda and Heki (2014) and Shinagawa et al. (2021) observed and modeled $f_oEs$ >20 MHz. However the phenomenon remains unpredictable, and its occurrence, intensity and spatial extent are not well constrained observationally, in particular over the oceans.

## 2. Long distance AIS links

The USCG AIS link dataset produced by stations in eastern USA contains almost 29 million links over three days. Most of the received data (>99%) are from ships within 300 km great circle distance of the USCG stations. The data were processed taking care to remove any repeated signals as well as AIS signals emitted from search and rescue aircraft. A histogram of observed AIS link distances is shown in Figure 1.

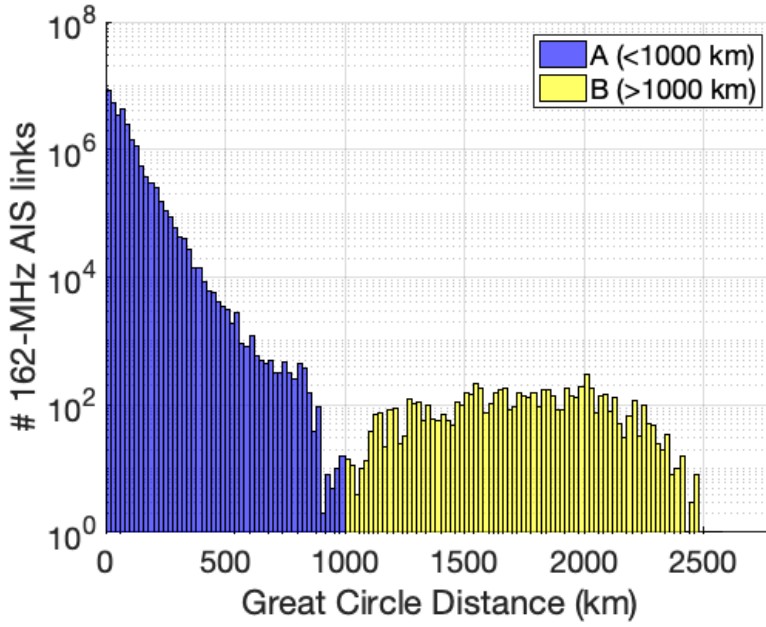

**Figure 1:** Histogram of AIS link great circle distances from USCG network in Eastern USA covering 13 -15 July 2021. Population A (<1000 km) might be explained by line-of-sight propagation, surface waves and tropospheric ducting, while Population B (>1000 km) is not predicted by those mechanisms. The longest reported link in our dataset covered a great-circle distance of 5453 km (not shown).

The ranges between a few hundred km and ~1000 km are likely caused by tropospheric refraction phenomena (e.g. ducting) and occur beyond normal line of sight, but are typically confined to paths within a few hundred km of the coastline. As can be seen, there is a distinct population of links observed above around 1000 km great circle distance, extending up to a maximum of 5453 km (6692 links >1000 km in total). A time series of these links is shown in Figure 2, with a representative day/night boundary for Bar Harbor, ME included for reference.

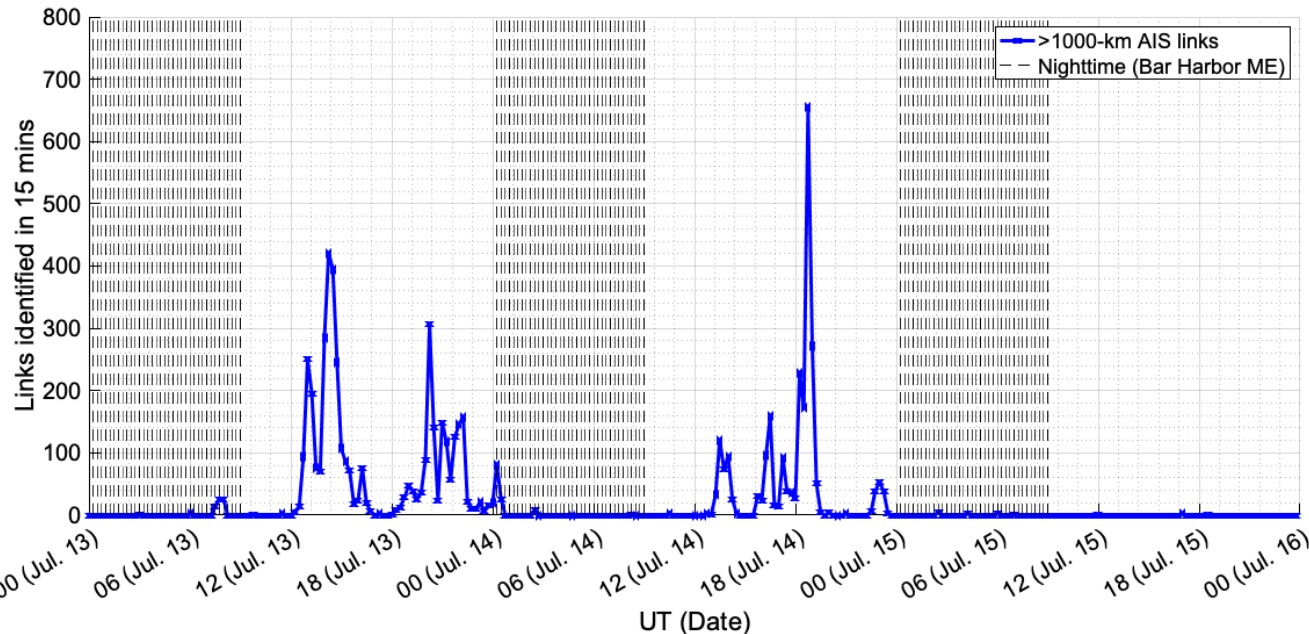

**Figure 2** shows AIS links received from ships at over 1000-km great circle distance from USCG stations between 13 – 15 July 2021. The results are binned in 15-minute increments. Nighttime at Bar Harbor, ME is indicated by black dashes.


Almost all (6677) of the long distance AIS links were detected on 13 and 14 July, with almost none seen at night or on 15 July. The maximum number of long-range links in a 15-minute window was 655, between 18:45-19:00 UT on 14 July, and the second-most was 421, between 14:15 – 14:30 UT on 13 July. Snapshots of these intervals are shown in Figure 3.

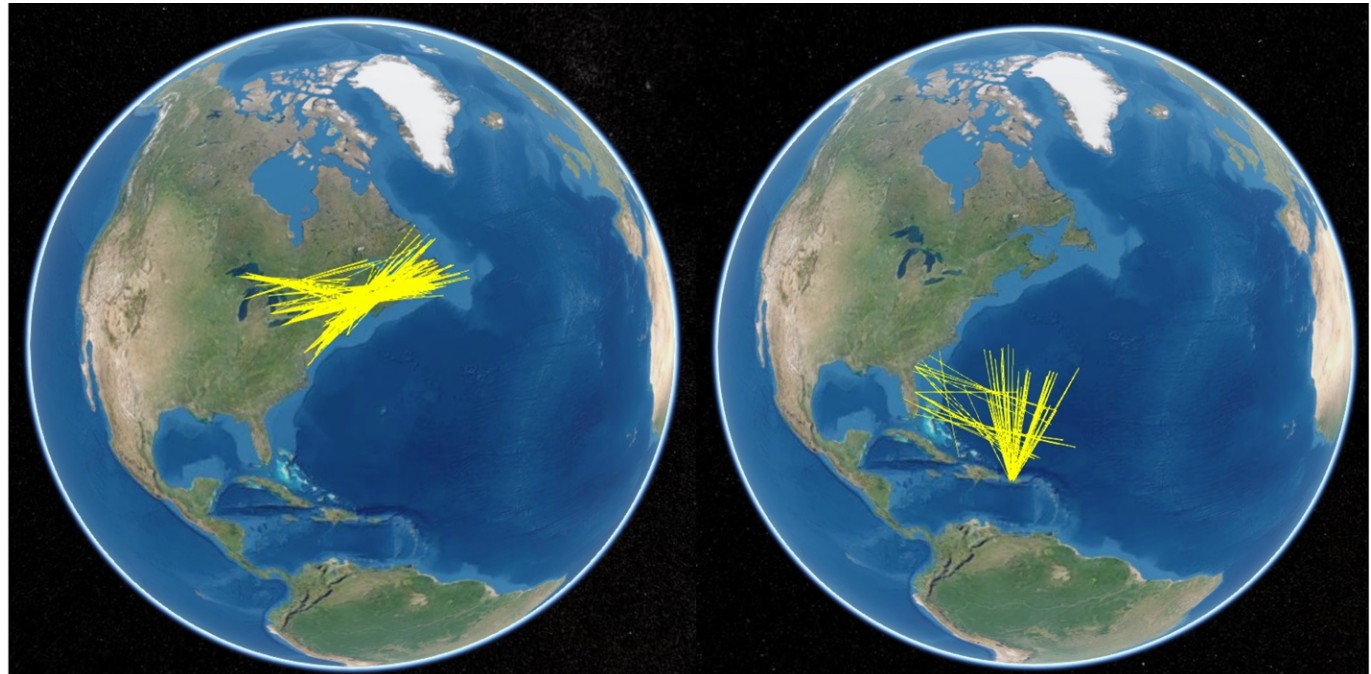

**Figure 3** shows the long-distance AIS links identified by USCG stations (left) between 18:45-19:00 UT on 14 July 2021 and (right) between 14:15 – 14:30 UT on 13 July 2021.

These snapshots indicate the long-distance AIS propagation is related to a spatially confined phenomenon, intermittently present over an area of hundreds or thousands of kilometers during daytime hours.


## 3. Data identifying tropospheric ducting

Tropospheric ducting is estimated using meteorological reanalysis temperature, pressure and humidity data. Here we computed
ducting using the European Centre for Medium-Range Weather Forecasts (ECMWF) 5th Generation Reanalysis (ERA5) described by Hersbach et al. (2018). These data were processed to compute the maximum strength of all RF/tropospheric ducts that appear at a single grid point and time. The maps of modified refractivity were derived from hourly forecast data using the full resolution 137 level temperature and specific humidity data provided on a regular Gaussian grid, at hourly resolution. Ducting maps at 19 UT on 14 July and 14 UT on 13 July are shown in Figure 4.


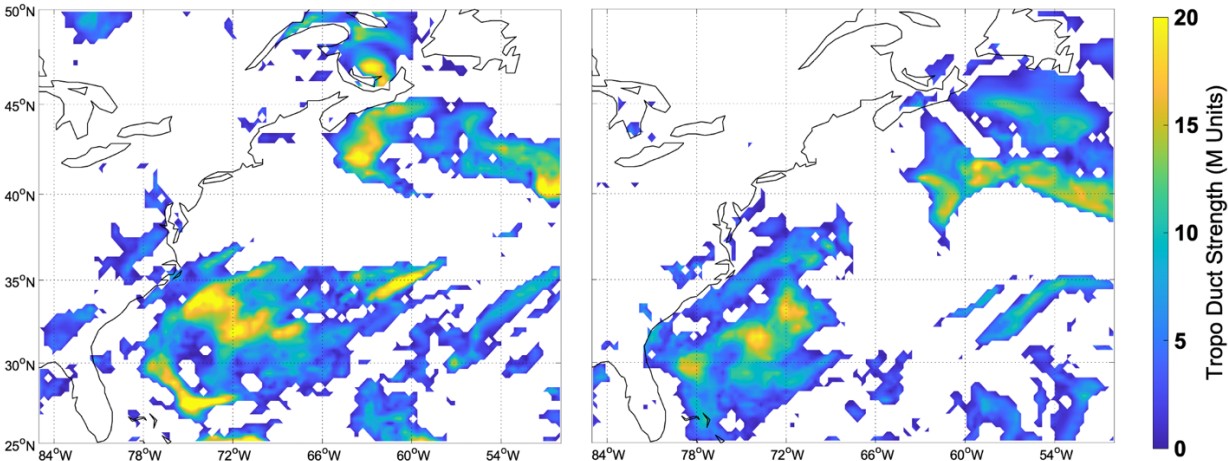

**Figure 4** shows tropospheric ducting maps of modified refractivity (in M units) based on temperatures, pressures and humidities in ERA5 reanalysis data. The left panel shows 19 UT on 14 July 2021, while the right panel shows 14 UT on 13 July 2021.

By showing the tropospheric ducting strength overlaid with the received position reports from land-based AIS receivers, one can see the correlations near shore of the tropospheric ducting that helps to extend the RF propagation range. It is also apparent that tropospheric ducting could not be responsible for the very long range AIS receptions that are the main focus of this publication. To illustrate this, a time series of the median nonzero tropospheric duct strength is shown in Figure 5.

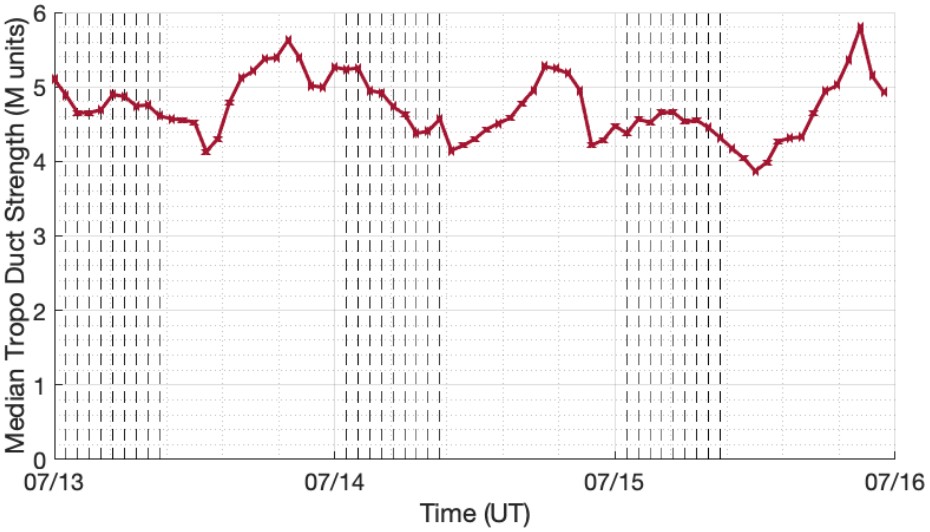

Figure 5: Median nonzero tropospheric duct strength between 25 – 50° N and 85 – 50° E between 13 – 15 July 2021. Nighttime at Bar Harbor, ME, is shown by black dashed lines.

As can be seen in Figure 5, tropospheric ducting peaks on 15 July 2021, when almost no long-distance AIS links were observed. Therefore a different explanation is needed for the >1000 km propagation observed on 162 MHz.

## 4. Data identifying sporadic-E

Data from groundbased Digisondes and from Constellation Observing System for Meteorology, Ionosphere, and Climate (COSMIC-2) radio occultations are used to identify the cause of the observed long-distance AIS links. Figure 6 shows

Digisonde peak sporadic-E plasma frequency ($f_oE_s$) from Millstone Hill, MA; Wallops Island, VA; and Ramey, Puerto Rico.

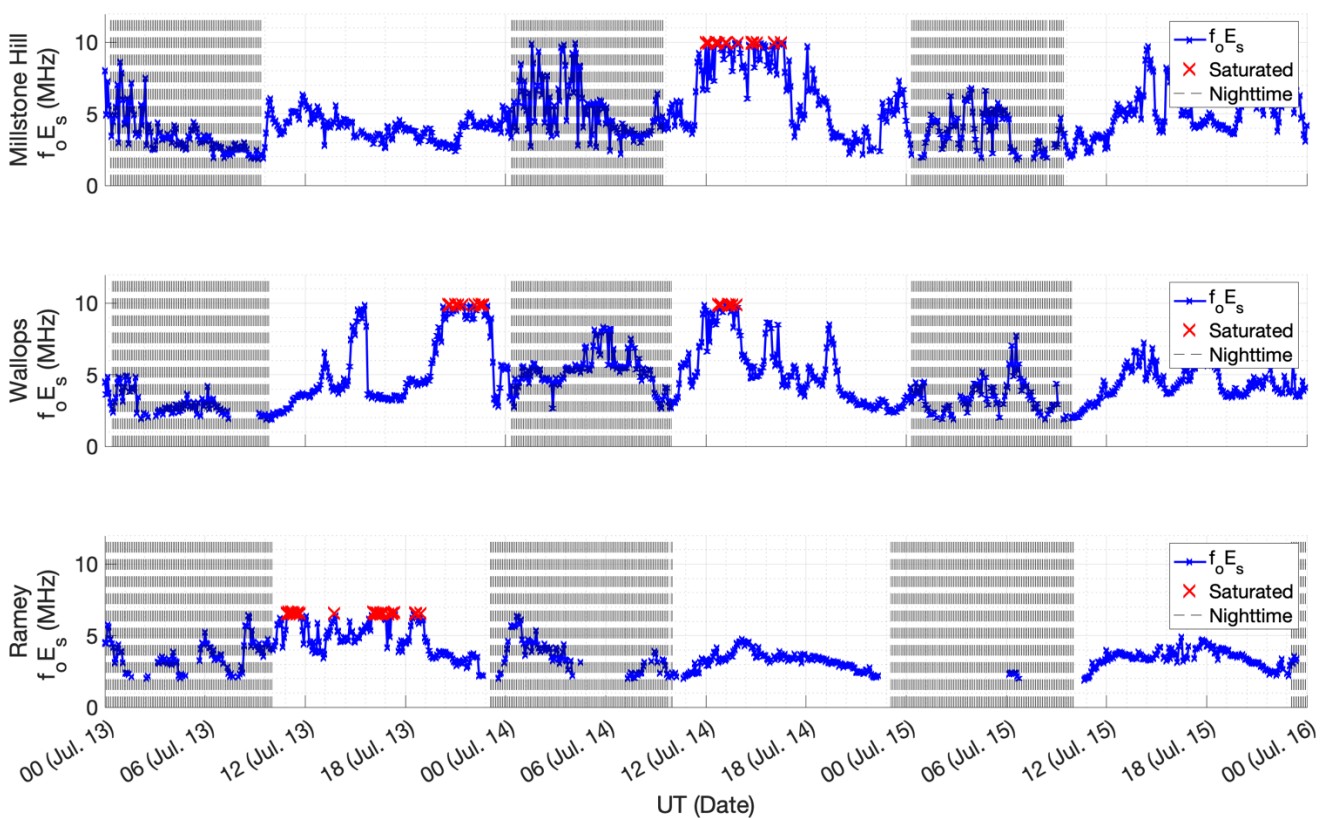

Figure 6: Autoscaled sporadic-E data from the Millstone Hill, Wallops and Ramey Digisondes. Saturated measurements where $f_oE_s$ exceeded the ionosonde's frequency range are shown in red. Local nighttime is indicated by black dashed lines.


The Digisonde $f_oE_s$ values saturate (i.e. they reach the maximum observable value of 10 MHz at Millstone and Wallops, or 6.5 MHz at Ramey) during the daytime at Wallops and Ramey on 13 July, and at Wallops and Millstone on 14 July. The data do

not saturate at any station either at night or at any time on 15 July. This is consistent with the trend seen in the long distance AIS link data.


Spatial maps of sporadic-E are produced by combining COSMIC-2 RO data with Digisonde measurements, using the $S_4$-based approach described by Yu et al. (2020). These maps are produced at three-hour cadence to increase the number of local observations in the region. Maps corresponding to the two maximum AIS link times are shown in Figure 7.

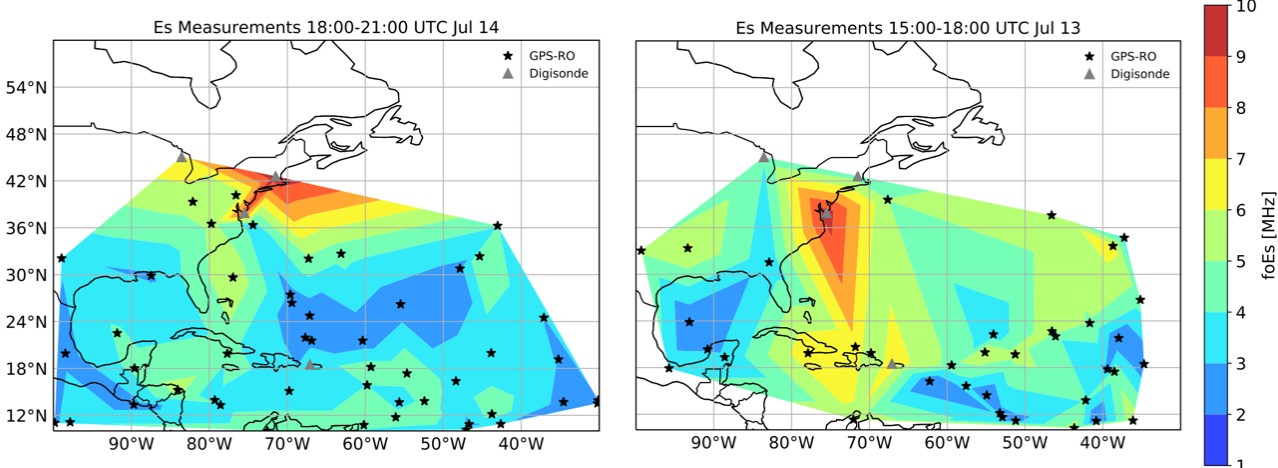

Figure 7 shows sporadic-E maps based on Digisonde and COSMIC-2 RO S4 data for 18-21 UT on 14 July 2021 and 15-18 UT on 13 July 2021.

Given the spatial and temporal limitations of the data coverage, the sporadic-E maps show remarkably good agreement with the AIS long distance link data shown in Figure 3. While the COSMIC-2 RO data is useful in identifying the presence or
absence of sporadic-E (Carmona et al., 2022), predicting $f_oE_s$ magnitudes from RO data is more difficult (Gooch et al., 2020). This is especially true of strong sporadic-E layers that tend to saturate the $S_4$ amplitude scintillation (Stambovsky et al., 2021; Yu et al., 2020). Therefore, we have higher confidence in the $f_oE_s$ estimates near Digisondes than in the locations driven strictly by RO data.

**5. Discussion and conclusions**

The analysis indicates that long distance 162 MHz AIS links observed in the western Atlantic region on 13 and 14 July 2021 are associated with sporadic-E layers. This is surprising given that signals at such high frequency are typically expected to

pass through the ionosphere. The available vertical incidence ionosonde data in the region saturate at or below 10 MHz, so it is not possible to determine the true value of $f_oE_s$ during these periods.


If we assume a layer height of 100-km, and take the shortest AIS paths within the anomalous distribution at around 1200-km (see Figure 1), we can estimate an angle of incidence ~9.5° (neglecting refraction and curvature). In this case the secant law (see Han, 1970 for details) implies $f_oE_s >= 27$ MHz, or $9 \times 10^{12}$ el. m$^3$. This is almost an order of magnitude larger than is typically expected, though there have been observations of $f_oE_s > 20$MHz (Chandra and Rastogi, 1975; Maeda and Heki, 2014).


There are several challenges inherent in using AIS data to identify long distance propagation. In some cases the system is subject to spoofing, either of the signals themselves or the GNSS signals they rely upon. There are also occasional transmission errors that still pass the checksum. Therefore accurate determination of long distance propagation requires the identification of multiple unique vessels over similar geometries. To avoid random errors, it is useful to observe multiple reports from a given ship over a few minutes; to observe consistency between the reported position, time, speed and course; and to cross-reference with other information, such as ports of call. We have applied these techniques in filtering the AIS data, but the removal of erroneous data remains an open challenge.


In summary, long distance AIS links appear to be a promising means of observing dense sporadic-E layers. The data have several advantages, notably high power, high density of users and high cadence due to internationally mandated usage, and coverage over the oceans. The USCG dataset used here is not routinely available, but the protocol is public and transponders are widespread on ships and ground stations. Therefore much of the infrastructure is already in place to develop a worldwide monitoring network. This study indicates skywave propagation due to sporadic-E layers is possible at higher frequencies than we were able to find previously reported in the literature, though we have not performed an exhaustive search.



## 6. Code availability

Data analysis code available here: https://github.com/alexchartier/sporadice

## 7. Data availability

Digisonde data can be retrieved from http://giro.uml.edu/didbase/scaled.php. COSMIC-2 data are available from https://doi.org/10.5065/t353-c093, while the sporadic-E maps for our study are available at https://zenodo.org/record/6977022#.YvJkKuzMJb8. Tropospheric ducting files are at: https://zenodo.org/record/7140002#.Yz3iqOzMJb8. The AIS data used here are not accessible to the public or research

community, but it may be possible to obtain them from USCG by request. We note that public repositories of other VHF radio link data exist that may be useful for independent validation of these findings (e.g. https://www.wsprnet.org/drupal/ and https://pskreporter.info/pskmap.html). The full set of >1000 km links can be viewed at https://youtu.be/AcNzM03zZP8.

## 8. Author Contribution

A.T. Chartier wrote the manuscript, produced figures 1-6, and submitted the paper. T.R. Hanley performed the AIS data analysis and first raised the possibility that these long-distance links were related to ionospheric phenomena. D.R. Emmons performed the ionosonde-GNSS sporadic E analysis and produced figure 7.

## 9. Competing Interests

The authors declare that they have no conflict of interest.

## 10. Acknowledgements

A.T. Chartier acknowledges support of NASA grant number 80NSSC21K1557.

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
