# Peer review of "Long Distance Propagation of 162-MHz Shipping Information Links Associated with Sporadic-E"

_Atmospheric Measurement Techniques, 2022_

## Author Response (AR1)

Reviewer Responses:

**Reviewer 1:**

Thanks for the review. We have uploaded the maps here:
https://zenodo.org/record/6977022#.YvJkKuzMJb8
We would like to add this link to the paper upon revision. We note the intense Es in the maps is driven by the Digisonde data (also shown independently earlier in the paper). There has been some success in linking GNSS sporadic E detections to other parameters (e.g. Yamazaki et al., 2022) but, as stated in the manuscript, the technique does not identify the true magnitude of these layers even compared to the 10 MHz-limited digisondes. We included these GNSS/Digisonde-based Es maps here as they show the spatial association to the anomalous AIS links, but we believe this association is driven by the availability of Digisonde data. The RO products (both TEC and S4) are not that sensitive to sporadic-E because the layers are so thin.

**Reviewer 2:**

Apologies for the delayed response. Co-author Hanley (the expert on tropo propagation in our group) was deployed on fieldwork until recently.

We have added tropo maps to the new draft, and have referenced the "More Miles on VHF" database. We did not see convincing evidence of >1000 km propagation caused by tropo there - in fact all the top 100 reported links are in prime sporadic-E season (May-August). However we have relaxed the wording to reflect that we believe it is unlikely, rather than impossible, that our >1000 km AIS links are from tropo ducting.

We have uploaded our tropo maps to Zenodo in case anyone wants to analyze the full set. It is quite clear (to us at least) that the >1000 km AIS links follow a completely separate pattern to those tropospheric ducts, both spatially and temporally. Conversely, the ionosonde foEs saturates (reaches maximum observable value) at times and locations corresponding to >1000 km AIS links.

We have addressed the technical points raised in the manuscript, and thank the reviewer for their insights.

---

## Referee Report (RR1)

Review on 1st revision of

Long Distance Propagation of 162-MHz Shipping
Information Links
Associated with Sporadic-E
Alex T. Chartier et al.

I appreciate the authors' work on improving the
manuscript since the initial submission. The paper
reads nicely, some doubts have been removed/solved and
very useful figure have been added.
Nevertheless I have a couple of points I'd like to
address in the following.

I strongly disagree with the statement in line 39 and
40. I suggest to either remove this sentence or
rephrase it in a way that there is indeed quite a
likelihood for tropospheric ducting for such distances,
which is the reason to investigate further in Sect. 3.
Perhaps this link (mmmonvhf) could also only be given
as the other references, but not in the text directly?

True, when you open the given link you'll first see the
claimed top distances for SporadicE propagation on
144MHz - marked as ES in the tickbox.

If you select the TR, which stands for tropospheric
ducting/scattering, you'll also see the list for this
propagation mode.
Here, most of the claims are OUTSIDE of the sporadicE
season, as it rather depends on the ducting situations,
which definitely not only occur in summer time,
especially for the mid-latitudes. Autumn and winter
time is actually more prominent, see e.g. reports for
Dec 2019 / Jan 2020, with ducts from UK to Cape Verde -
even on 432MHz! Such distances for 144MHz would only be

explainable by at least 2x Es plus tropo assistance...
but certainly not in January and not for 432MHz.
https://qrznow.com/432-mhz-world-tropo-record-extended-
even-further-to-4644-kms/

L 44 :   e.g Thomas  -> e.g. Thomas

L 94 : good to see it doesn't match to the Es paths
Fig. 3

Fig. 5 : thanks for adding this also, perhaps a bit
large, and/or should be scaled (y-axis) differently

L 110 : I'd suggest to refer here to the Figure 2 to
highlight the disagreement. Alternatively overlay both
figures, but this is certainly more of an effort.
Furthermore it would be interessting/useful to have a
value for a pronounced tropo duct strength (M units),
to judge how far it is off to the observed values that
.

L 142 : "This is surprising given..." Well, not really
surprising... perhaps remarkable?...

L 146 : 100-km -> 100 km , 1200-km -> 1200 km
L 149 : 20MHz -> 20 MHz

---

## Author Response (AR2)

Thanks for the comments. I have made the requested changes.

I would be curious to know how the MMMonVHF and others determine the propagation mode. My guess is they are making some association with other factors (e.g. moon location, time of day/year etc) and maybe also accounting for signal-to-noise and other parameters. It would be useful to see a description of the classifier.